# Deterministic realization of collective measurements via photonic quantum walks

Zhibo Hou[1,2], Jun-Feng Tang[1,2], Jiangwei Shang[3,4], Huangjun Zhu[5,6,7,8,9], Jian Li[10,11], Yuan Yuan[1,2], Kang-Da Wu[1,2], Guo-Yong Xiang[1,2], Chuan-Feng Li[1,2] & Guang-Can Guo[1,2]

Collective measurements on identically prepared quantum systems can extract more information than local measurements, thereby enhancing information-processing efficiency. Although this nonclassical phenomenon has been known for two decades, it has remained a challenging task to demonstrate the advantage of collective measurements in experiments. Here, we introduce a general recipe for performing deterministic collective measurements on two identically prepared qubits based on quantum walks. Using photonic quantum walks, we realize experimentally an optimized collective measurement with fidelity 0.9946 without post selection. As an application, we achieve the highest tomographic efficiency in qubit state tomography to date. Our work offers an effective recipe for beating the precision limit of local measurements in quantum state tomography and metrology. In addition, our study opens an avenue for harvesting the power of collective measurements in quantum information-processing and for exploring the intriguing physics behind this power.

[1] Key Laboratory of Quantum Information, University of Science and Technology of China, CAS, Hefei 230026, P. R. China. [2] Synergetic Innovation Center of Quantum Information and Quantum Physics, University of Science and Technology of China, Hefei 230026, P. R. China. [3] Naturwissenschaftlich-Technische Fakultät, Universität Siegen, Siegen 57068, Germany. [4] Beijing Key Laboratory of Nanophotonics and Ultrafine Optoelectronic Systems, School of Physics, Beijing Institute of Technology, Beijing 100081, China. [5] Institute for Theoretical Physics, University of Cologne, Cologne 50937, Germany. [6] Department of Physics and Center for Field Theory and Particle Physics, Fudan University, Shanghai 200433, China. [7] Institute for Nanoelectronic Devices and Quantum Computing, Fudan University, Shanghai 200433, China. [8] State Key Laboratory of Surface Physics, Fudan University, Shanghai 200433, China. [9] Collaborative Innovation Center of Advanced Microstructures, Nanjing 210093, China. [10] Institute of Signal Processing Transmission, Nanjing University of Posts and Telecommunications, Nanjing 210003, China. [11] Key Lab of Broadband Wireless Communication and Sensor Network Technology, Nanjing University of Posts and Telecommunications, Ministry of Education, Nanjing 210003, China. Correspondence and requests for materials should be addressed to H.Z. (email: zhuhuangjun@fudan.edu.cn) or to G.-Y.X. (email: gyxiang@ustc.edu.cn)

Quantum measurements are the key for extracting information from quantum systems and for connecting the quantum world with the classical world. Understanding the power and limitation of measurements is of paramount importance not only to foundational studies, but also to many applications, such as quantum tomography, metrology, and communication[1–8]. An intriguing phenomenon predicted by quantum theory is that collective measurements on identically prepared quantum systems may extract more information than local measurements on individual systems, thereby leading to higher tomographic efficiency and precision[9–14]. The significance of collective measurements for multiparameter quantum metrology was also recognized recently[15,16]. This nonclassical phenomenon is owing to entanglement in the quantum measurements instead of quantum states. It is closely tied to the phenomenon of "nonlocality without entanglement"[17]. In addition, collective measurements are very useful in numerous other tasks, such as distilling entanglement[18], enhancing nonlocal correlations[19], and detecting quantum change point[20]. However, demonstrating the advantage of collective measurements in experiments has remained a daunting task. This is because most optimized protocols entail generalized entangling measurements on many identically prepared quantum systems, which are very difficult to realize deterministically.

Here we introduce a general method for performing deterministic collective measurements on two identically prepared qubits based on quantum walks, which extends the method for performing generalized measurements on a single qubit only[21–23]. By devising photonic quantum walks, we realize experimentally a highly efficient collective measurement highlighted in refs. [11,13,14]. As an application, we realize, for the first time, qubit state tomography with deterministic collective measurements. The protocol we implemented is significantly more efficient than local measurements commonly employed in most experiments. Moreover, it can achieve near-optimal performance over all two-copy collective measurements with respect to various figures of merit without using adaptive measurements. Such high efficiency demonstrates the main advantage of collective measurements over separable measurements. Here, we encode the two qubits in the two degrees of freedom of a single photon[24–27], but our method for performing collective measurements can be generalized to two-photon two-qubit states by combining the technique of quantum joining[28] or teleportation[29].

## Results

**Optimized collective measurements.** In quantum theory, a measurement is usually represented by a positive-operator-valued measure (POVM), which is composed of a set of positive operators that sum up to the identity. In traditional quantum information-processing, measurements are performed on individual quantum systems one by one, which often cannot extract information efficiently. Fortunately, quantum theory allows us to perform collective measurements on identically prepared quantum systems in a way that has no classical analog, as illustrated in Fig. 1.

In the case of a qubit, a special two-copy collective POVM was highlighted in refs.[11,13,14], which consists of five POVM elements,

$$E_j = \frac{3}{4}\left(|\psi_j\rangle\langle\psi_j|\right)^{\otimes 2}, \qquad E_5 = |\Psi_-\rangle\langle\Psi_-|, \qquad (1)$$

where $|\Psi_-\rangle = \frac{1}{\sqrt{2}}(|01\rangle - |10\rangle)$ is the singlet, which is maximally entangled, and $|\psi_j\rangle$ for $j = 1, 2, 3, 4$ are qubit states that form a symmetric informationally complete POVM (SIC-POVM), that is, $\left|\langle\psi_j|\psi_k\rangle\right|^2 = (2\delta_{jk} + 1)/3$[30,31]. Geometrically, the Bloch vectors of the four states $|\psi_j\rangle$ form a regular tetrahedron inside the Bloch sphere. For concreteness, here we choose

$$|\psi_1\rangle = |0\rangle, \qquad |\psi_2\rangle = \frac{1}{\sqrt{3}}\left(|0\rangle + \sqrt{2}|1\rangle\right),$$
$$|\psi_3\rangle = \frac{1}{\sqrt{3}}\left(|0\rangle + e^{\frac{2\pi i}{3}}\sqrt{2}|1\rangle\right), \qquad (2)$$
$$|\psi_4\rangle = \frac{1}{\sqrt{3}}\left(|0\rangle + e^{-\frac{2\pi i}{3}}\sqrt{2}|1\rangle\right).$$

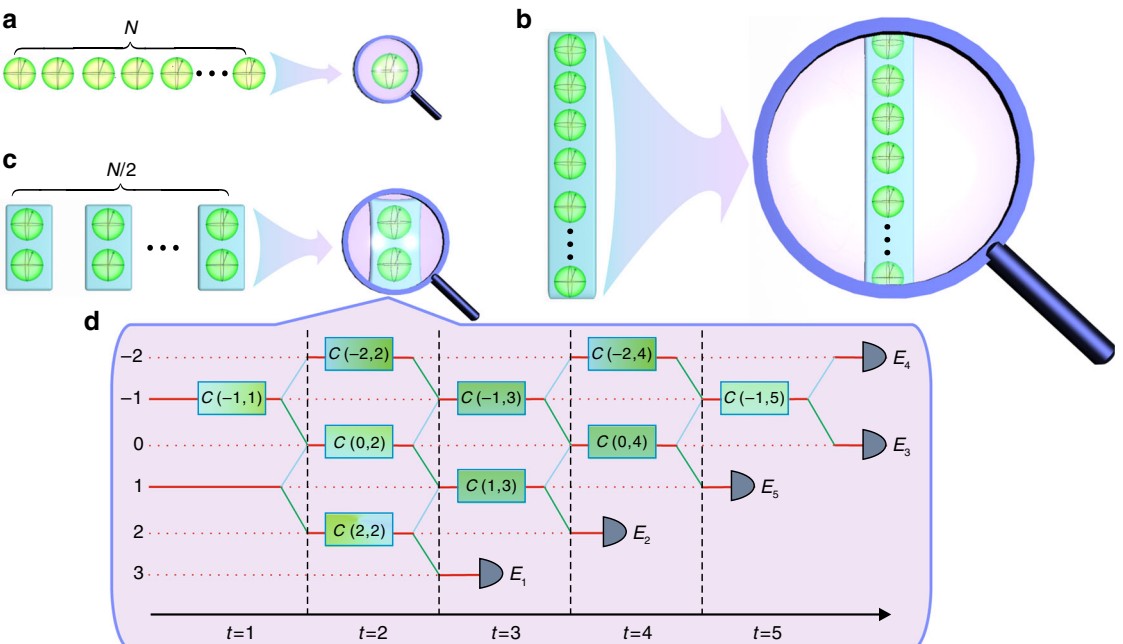

**Fig. 1** Individual and collective measurements. **a** Repeated individual measurements. **b** Single $N$-copy collective measurement. **c** Repeated two-copy collective measurements. **d** Realization of the collective SIC-POVM defined in Eqs. (1) and (2) using five-step quantum walks. The coin qubit and the walker in positions 1 and −1 are taken as the two-qubit system of interest, whereas the other positions of the walker act as an ancilla. Site-dependent coin operators $C(x, t)$ are specified in the Methods section. Five detectors $E_1$ to $E_5$ correspond to the five outcomes of the collective SIC-POVM

The POVM defined by Eqs. (1) and (2) is referred to as the collective SIC-POVM henceforth. If this POVM is performed on the two-copy state $\rho^{\otimes 2}$, then the probability of obtaining outcome $j$ is given by $p_j = \mathrm{tr}(\rho^{\otimes 2} E_j)$.

The collective SIC-POVM is distinguished because it is optimal in extracting information from a pair of identical qubits[9,11]. It is universally Fisher symmetric in the sense of providing uniform and maximal Fisher information on all parameters that characterize the quantum states of interest[13,14,32]. Moreover, it is unique such POVM with no more than five outcomes. Consequently, the collective SIC-POVM is significantly more efficient than any local measurement in many quantum information-processing tasks, including tomography and metrology. Moreover, its high tomographic efficiency is achieved without using adaptive measurements, which is impossible for local measurements. As far as two-copy collective measurements are concerned, surprisingly, more entangled measurements, such as the Bell measurements, cannot lead to higher efficiency. Although multi-copy (say three-copy) collective measurements can further improve the efficiency, the improvement is not so significant[13,14].

**Realization of the collective SIC-POVM via quantum walks.** Recently, discrete quantum walks were proposed as a recipe for implementing general POVMs on a single qubit[21], which have been demonstrated in experiments[22,23]. In a one-dimensional discrete quantum walk, the system state is characterized by two degrees of freedom $|x, c\rangle$, where $x = \ldots, -1, 0, 1, \ldots$ denotes the walker position, and $c = 0, 1$ represents the coin state. The dynamics of each step is described by a unitary transformation of the form $U(t) = T C(t)$, where

$$T = \sum_x |x + 1, 0\rangle\langle x, 0| + |x - 1, 1\rangle\langle x, 1| \quad (3)$$

is the conditional translation operator, and $C(t) = \sum_x |x\rangle\langle x| \otimes C(x, t)$ with $C(x, t)$ being site-dependent coin operators. A general POVM on a qubit can be realized by engineering the coin operators $C(x, t)$ followed by measuring the walker position after certain steps. However, little is known in the literature on realizing POVMs on higher-dimensional systems. Here, we propose a general method for extending the capabilities of quantum walks. For concreteness, we illustrate our approach with the collective SIC-POVM.

To realize the collective SIC-POVM using quantum walks, the coin qubit and the walker in positions 1 and −1 are taken as the two-qubit system of interest, whereas the other positions of the walker act as an ancilla. With this choice, the collective SIC-POVM can be realized with five-step quantum walks, as illustrated in Fig. 1d and discussed in more details in Supplementary Note 1. Here, the nontrivial coin operators $C(x, t)$ are specified in the Methods section. The five detectors $E_1$ to $E_5$ marked in the figure correspond to the five POVM elements specified in Eqs. (1) and (2). Moreover, this proposal can be implemented using photonic quantum walks, as illustrated in Fig. 2 (see also Supplementary Fig. 1).

**Experimental setup.** The experimental setup for realizing the collective SIC-POVM and its application in quantum state tomography is presented in Fig. 2. The setup is composed of two modules designed for two-copy state-preparation and collective measurements, respectively.

The two-copy collective measurement module performs the collective SIC-POVM based on quantum walks, as illustrated in Fig. 1d (cf. Supplementary Fig. 1). Here the conditional translation operator $T$ is realized by interferometrically stable beam displacers (BDs)[33–36], which displace the component with horizontal polarization (H) away from the component with vertical polarization (V). The coin operators $C(x, t)$ are realized by suitable combinations of half wave plates (HWPs) and quarter wave plates (QWPs), with rotation angles specified in the table embedded in Fig. 2.

In the two-copy state-preparation module, we first prepare copy 1 in the path degree of freedom, i.e., the walker qubit encoded in positions 1 and −1 (see **a** in Fig. 2). A pair of 1-mm-

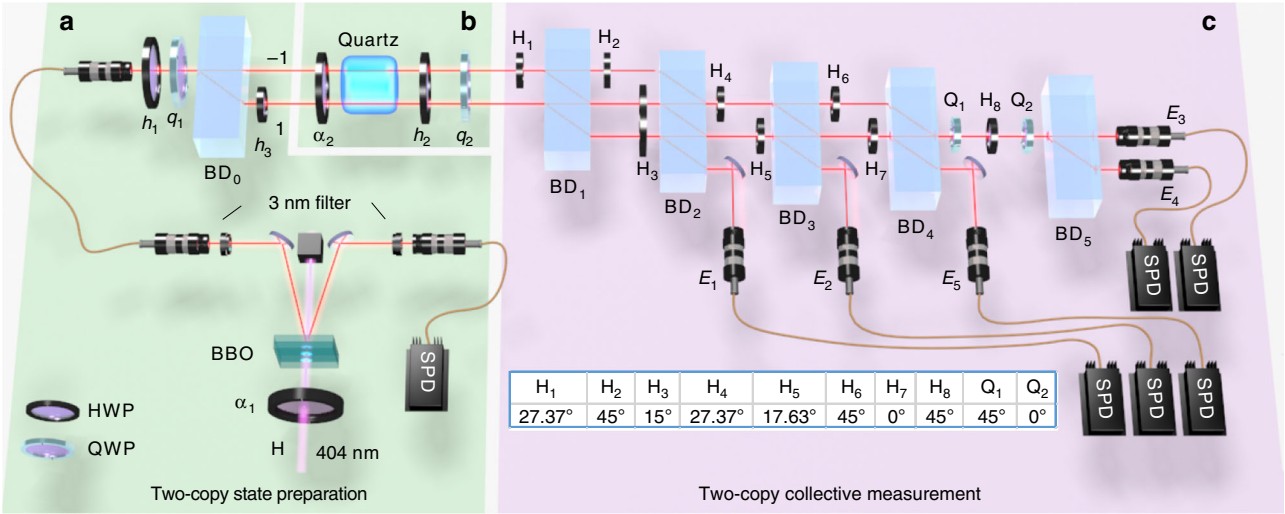

**Fig. 2** Experimental setup for realizing the collective SIC-POVM. The setup consists of two modules designed for two-copy state preparation (**a**, **b**) and two-copy collective measurement **c**, respectively. In the two-copy state-preparation module, **a** prepares the first copy (walker qubit) in the path degree of freedom; **b** prepares the second copy (coin qubit) in the polarization degree of freedom. The two-copy collective measurement module (**c**) performs the collective SIC-POVM via photonic quantum walks as illustrated in Fig. 1d. Here beam displacers (BDs) are used to realize the conditional translation operator $T$. Combinations of half wave plates (HWPs) and quarter wave plates (QWPs) with rotation angles specified in the embedded table are used to realize site-dependent coin operators $C(x, t)$. Five single-photon detectors (SPDs) $E_1$ to $E_5$ correspond to the five outcomes of the collective SIC-POVM

| $H_1$ | $H_2$ | $H_3$ | $H_4$ | $H_5$ | $H_6$ | $H_7$ | $H_8$ | $Q_1$ | $Q_2$ |
|---|---|---|---|---|---|---|---|---|---|
| 27.37° | 45° | 15° | 27.37° | 17.63° | 45° | 0° | 45° | 45° | 0° |

long BBO crystals with optical axes perpendicular to each other, cut for type-1 phase-matched spontaneous parametric down-conversion (SPDC) process, is pumped by a 40-mW H-polarized beam at 404 nm. The polarization state of the beam is prepared as $\cos 2\alpha_1 |H\rangle + \sin 2\alpha_1 |V\rangle$ when the deviation angle of the HWP at 404 nm is set at $\alpha_1$. After the SPDC process, a pair of photons with wave length $\lambda = 808$ nm is created in the state of $\sin 2\alpha_1 |HH\rangle + \cos 2\alpha_1 |VV\rangle$[37]. The two photons pass through two interference filters whose FWHM (full width at half maximum) is 3 nm, resulting in a coherence length of $270\lambda$. One photon is detected by a single-photon detector acting as a trigger. After tracing out this photon, the other photon is prepared in the state $\sin^2 2\alpha_1 |H\rangle\langle H| + \cos^2 2\alpha_1 |V\rangle\langle V|$, whose purity is controlled by $\alpha_1$. Two HWPs (not shown in Fig. 2) at the input and output ports of the single-mode fiber are used to maintain the polarization state of the photon. After passing a

HWP and a QWP with deviation angles $h_1$, $q_1$, the photon is prepared in the desired state $\rho$. To encode the polarization state into the path degree of freedom, $BD_0$ is used to displace the H-component into path 1, which is 4-mm away from the V-component in path $-1$; then a HWP with deviation angle $h_3 = 45°$ is placed in path 1. The resulting photon is described by the state $\rho \otimes |V\rangle\langle V|$.

Then we encode the second copy of $\rho$ into the polarization degree of freedom (coin qubit) using two HWPs, a quartz crystal with a decoherence length of $385\lambda$, and a QWP (see **b** in Fig. 2). The first HWP with rotation angle $\alpha_2$ and the quartz crystal prepare the polarization state $\sin^2 2\alpha_2 |H\rangle\langle H| + \cos^2 2\alpha_2 |V\rangle\langle V|$ with desired purity. Then the direction of the Bloch vector of the polarization state is adjusted by a HWP and a QWP with deviation angles $h_2$ and $q_2$. In this way, we can prepare the desired two-copy state $\rho \otimes \rho$, the first copy of which is encoded in the path degree of freedom, whereas the second one in the polarization degree of freedom.

The two-copy state-preparation module described above is capable of preparing any two-copy state (see Supplementary Note 2 for more details). Next, the two-copy state $\rho \otimes \rho$ is sent into the two-copy collective measurement module, which performs the collective SIC-POVM based on quantum walks, as described before. It is worth pointing out that the collective SIC-POVM can also be applied to measure arbitrary two-qubit states, although we focus on two-copy qubit states in this work.

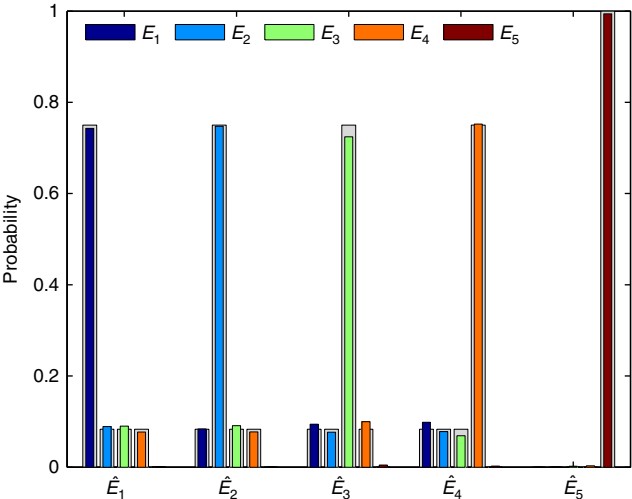

**Fig. 3** Experimental verification of the collective SIC-POVM realized. Here each $\hat{E}_i$ for $i = 1, 2, 3, 4, 5$ denotes an input state, which corresponds to the POVM element $E_i$ of the collective SIC-POVM after normalization. Each input state is prepared and measured 100,000 times. The frequencies of obtaining the five outcomes are plotted using different colors; here the error bars are too small to be visible. For comparison, the probabilities in the ideal scenario are plotted in gray shadow

**Verification and tomography of the collective SIC-POVM.** To verify the experimental implementation of the collective SIC-POVM, we took the conventional method of measuring the probability distributions after preparing the input states as normalized POVM elements, i.e., $\hat{E}_i = E_i/\text{tr}(E_i)$ for $i = 1, \ldots, 5$. These input states can be prepared by choosing proper rotation angles $\alpha_1, h_1, q_1, h_3, \alpha_2, h_2, q_2$ as specified in Supplementary Table 1. The measurement probability distributions are shown in Fig. 3, which agree very well with the theoretical prediction.

To accurately characterize the POVM elements that were actually realized, we then performed quantum measurement tomography. Overall, 36 input states, the tensor products of the six eigenstates of three Pauli operators, were prepared and sent to the collective measurement module, with each setting repeated 35,000 times. Then the five POVM elements were estimated from the measurement statistics using the maximum likelihood method developed in ref.[38]. The fidelities of the five POVM

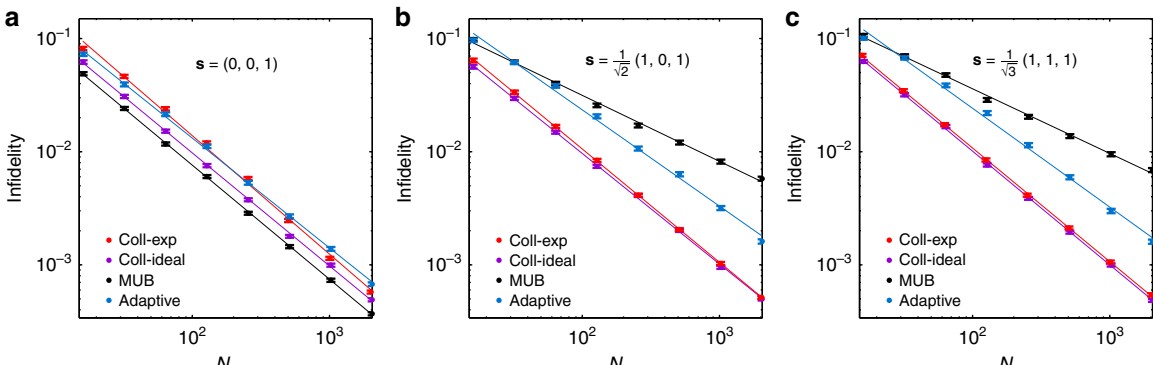

**Fig. 4** Scaling of the mean infidelity in quantum state tomography with the collective SIC-POVM. Both experimental (coll-exp) and simulation (coll-ideal) results are plotted for the collective SIC-POVM. The performances of MUB and two-step adaptive measurements (simulation) are shown for comparison. **a**, **b** and **c** correspond to the tomography of three pure states with Bloch vectors **s** as specified; N is the sample size, ranging from 16 to 2048. Each data point is the average of 1000 repetitions, and the error bar denotes the standard deviation

elements estimated are 0.9991 ± 0.0001, 0.9979 ± 0.0007, 0.9870 ± 0.0008, 0.9927 ± 0.0002 and 0.9961 ± 0.0002, respectively; the overall fidelity of the POVM (cf. the Methods section) is 0.9946 ± 0.0002. Here, the error bars denote the standard deviations of 100 simulations from Poisson statistics. Such high fidelities demonstrate that the collective SIC-POVM was realized with very high quality. Detailed information about the five reconstructed POVM elements can be found in Supplementary Note 3 and Supplementary Fig. 2.

**Quantum state tomography with the collective SIC-POVM.** The experimental realization of the collective SIC-POVM enables us to achieve unprecedented efficiency in quantum state tomography. In this section we demonstrate the tomographic significance of the collective SIC-POVM and the power of collective measurements.

In the first experiment, we investigated the scaling of the mean infidelity $1 - F$ achieved by the collective SIC-POVM with the sample size $N$ (the number of copies of the state available for tomography). Three pure states with Bloch vectors $(0, 0, 1)$, $\frac{1}{\sqrt{2}}(1, 0, 1)$, and $\frac{1}{\sqrt{3}}(1, 1, 1)$ were considered (see Supplementary Note 4 and Supplementary Fig. 3 for additional results on mixed states). In each case, the probabilities of obtaining the outcomes of the collective SIC-POVM were estimated from frequencies of

repeated measurements, from which we reconstructed the original state using the maximum likelihood method[4]; see Supplementary Note 5 and Supplementary Fig. 4.

The experimental result and simulation result are shown in Fig. 4. Also shown as benchmarks are the simulation results on two popular alternative schemes: one based on mutually unbiased bases (MUB) for a qubit[39–42] and the other based on two-step adaptive measurements proposed in ref. [43] (cf. refs.[44–46]). The experimental result agrees very well with the theoretical predication[14] and numerical simulation. The efficiency of the collective SIC-POVM is almost independent of the input state; the infidelity approximately scales as $O(1/N)$ for all states investigated (cf. Supplementary Table 2). This high efficiency is tied to the fact that the probability of obtaining the outcome $E_5$ in Eq. (1) is very sensitive to the purity of the input state, so that the purity can be estimated very accurately. By contrast, the scaling behavior is much worse for MUB except when the input state aligns with one of the POVM elements, which corresponds to "known state tomography"[43]. This is because the infidelity is very sensitive to inaccurate estimation of the purity, which is unavoidable for a fixed individual measurement. For a generic pure state, the infidelity achieved by the collective SIC-POVM for $N = 2048$ is ~ 12 (three) times smaller than that achieved by MUB (local adaptive measurements). The advantage of the

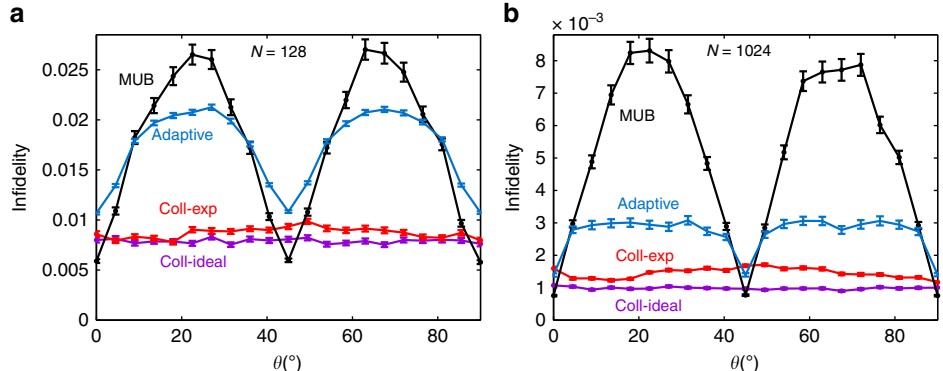

**Fig. 5** Mean infidelities achieved by the collective SIC-POVM in estimating a family of pure states. These pure states have the form $|\psi(\theta)\rangle = \sin\theta|0\rangle + \cos\theta|1\rangle$. The performances of MUB and two-step adaptive measurements (simulation) are shown for comparison. The sample size is $N = 128$ in **a** and $N = 1024$ in **b**. Each data point is the average of 1000 repetitions, and the error bar denotes the standard deviation

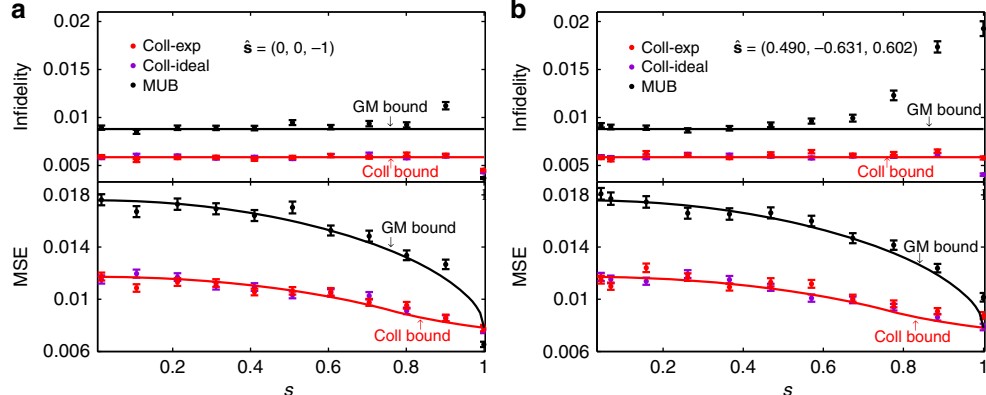

**Fig. 6** Performance of the collective SIC-POVM in the tomography of mixed qubit states. Two families of mixed states with Bloch vector directions specified in **a** and **b** are considered. The figures of merit are chosen as the mean infidelity and MSE. Also shown for comparison are the performance of MUB (simulation) as well as the Gill-Massar (GM) bound[13,44,47] and a collective (coll) bound[13,14] (see the Methods section). Here $\hat{s}$ and $s$ denote the direction and length of the Bloch vector; the sample size is $N = 256$; each data point is the average of 1000 repetitions, and the error bar denotes the standard deviation

collective SIC-POVM becomes more significant as the sample size increases.

In the second experiment, we investigated the mean infidelity achieved by the collective SIC-POVM when the input states have the form $|\psi(\theta)\rangle = \sin\theta|0\rangle + \cos\theta|1\rangle$ with $\theta$ ranging from 0 to $\pi/2$. Here $N$ is chosen to be 128 (that is, 64 pairs) or 1024 (512 pairs). The result shown in Fig. 5 further demonstrates that the efficiency of the collective SIC-POVM is almost independent of the input state. In addition, the infidelity in the worst scenario is much smaller than that achieved by MUB and local adaptive measurements. As in the first experiment, the advantage of the collective SIC-POVM becomes more significant when $N$ increases.

In the third experiment, we considered two families of mixed states $\rho = \frac{1}{2}(I + \mathbf{s} \cdot \sigma)$ with Bloch vectors along $\hat{\mathbf{s}} = (0, 0, -1)$ and $\hat{\mathbf{s}} = (0.490, -0.631, 0.602)$, respectively, and with $s$ ranging from 0 to 1. The sample size $N$ is chosen to be 256; both the mean infidelity and mean square error (MSE) are considered as figures of merit. The experimental result is shown in Fig. 6. The mean infidelity achieved by the collective SIC-POVM is not only smaller than that by MUB, but also smaller than the Gill-Massar (GM) bound[13,44,47], which constrains the performance of any local measurement, even with adaptive choices. Moreover, the mean infidelity approximately saturates a bound derived in refs. [13,14], which represents the best performance that can be achieved by two-copy collective measurements; cf. the Methods section. In addition, the collective SIC-POVM is also nearly optimal with respect to the MSE for all states. Remarkably, such high efficiency is achieved without any adaptive measurement.

## Discussion

In summary, we introduced a general method for implementing deterministic collective measurements on two identically pre-pared qubits based on quantum walks. Using photonic quantum walks, we then realized experimentally the collective SIC-POVM with very high quality and thereby achieved unprecedented high efficiency in qubit state tomography. The collective SIC-POVM we realized is significantly more efficient than any local mea-surement. It improves the scaling of the mean infidelity in the worse scenario from $O(1/\sqrt{N})$ to $O(1/N)$. Moreover, it is nearly optimal over all two-copy collective measurements with respect to various figures of merit, including the mean infidelity and MSE, although no adaptive measurement is required. This high effi-ciency manifests the primary advantage of collective measure-ments over separable measurements.

Our work demonstrated a truly nonclassical phenomenon that is owing to entanglement in quantum measurements instead of quantum states. Moreover, it offers an effective recipe for exceeding the precision limit of local measurements in quantum state tomography. Similar idea can readily be applied to enhance the precision in multiparameter quantum metrology. For example, in the joint estimation of phase and phase diffusion, it was shown in ref. [15] that collective measurements can lead to higher precision than local measurements. Recently, this prediction was verified in a proof-of-principle experiment based on probabilistic Bell measurements[16]. Our technique for implementing deterministic collective measurements may help demonstrate the practical advantage of collective measurements in quantum metrology. More generally, our work opens an avenue for exploring the power of collective measurements in quantum information processing. In the future, it would be desirable to extend our approach to realize multi-copy collective measurements on qubits and systems of higher dimensions.

## Methods

**Coin operators for realizing the collective SIC-POVM.** Here, we present the coin operators that appear in Fig. 1d; see Supplementary Note 1 for more details.

$$C(-1,1) = \tfrac{1}{\sqrt{3}}\begin{pmatrix} 1 & \sqrt{2} \\ \sqrt{2} & -1 \end{pmatrix}, \quad C(-2,2) = \begin{pmatrix} 0 & 1 \\ 1 & 0 \end{pmatrix},$$

$$C(0,2) = \tfrac{1}{2}\begin{pmatrix} \sqrt{3} & 1 \\ 1 & -\sqrt{3} \end{pmatrix}, \quad C(1,3) = \tfrac{1}{\sqrt{3}}\begin{pmatrix} \sqrt{2} & 1 \\ 1 & -\sqrt{2} \end{pmatrix},$$

$$C(0,4) = \begin{pmatrix} 1 & 0 \\ 0 & -1 \end{pmatrix}, \quad C(-1,5) = \tfrac{1}{2}\begin{pmatrix} 1-i & 1+i \\ -1+i & 1+i \end{pmatrix},$$

$$C(2,2) = C(0,2), \quad C(-1,3) = C(-1,1),$$

$$C(-2,4) = C(-2,2). \tag{4}$$

**Fidelity between two POVMs.** Consider two POVMs $\{E_j\}_{j=1}^M$ and $\{E_{j'}\}_{j=1}^M$ on a $d$-dimensional Hilbert space with the same number of elements, where $E_{j'}$ is the counterpart of $E_j$ (for example, one is the experimental realization of the other). Construct two normalized quantum states as $\sigma = \frac{1}{d}\sum_{j=1}^M E_j \otimes (|j\rangle\langle j|)$ and $\sigma' = \frac{1}{d}\sum_{j=1}^M E_{j'} \otimes (|j\rangle\langle j|)$, where $|j\rangle$ form an orthonormal basis for an ancilla system. The fidelity between the two POVMs $\{E_j\}_{j=1}^M$ and $\{E_{j'}\}_{j=1}^M$ is defined as the fidelity between the two states $\sigma$ and $\sigma'$,

$$F(\sigma, \sigma') := \left(\mathrm{tr}\sqrt{\sqrt{\sigma}\sigma'\sqrt{\sigma}}\right)^2 = \left(\sum_{j=1}^M w_j \sqrt{F_j}\right)^2, \tag{5}$$

where $w_j = \frac{\sqrt{\mathrm{tr}(E_j)\mathrm{tr}(E_{j'})}}{d}$, and $F_j = F\left(\frac{E_j}{\mathrm{tr}(E_j)}, \frac{E_{j'}}{\mathrm{tr}(E_{j'})}\right)$ is the fidelity between the two normalized POVM elements $\frac{E_j}{\mathrm{tr}(E_j)}$ and $\frac{E_{j'}}{\mathrm{tr}(E_{j'})}$.

**GM bounds and collective bounds.** In quantum state tomography with individual measurements (including local adaptive measurements), the precision achievable is constrained by the GM bound[13,44,47] (see also ref. [48]). In the case of a qubit, the GM bound is $\frac{9}{4N}$ when the figure of merit is the mean infidelity (approximately equal to the mean square Bures distance), where $N$ is the sample size (assuming $N$ is not too small). When the figure of merit is the MSE $\mathbb{E}\,\mathrm{tr}[(\hat\rho - \rho)^2]$, the GM bound is $\frac{(2+\sqrt{1-s^2})^2}{2N}$, where $s$ is the length of the Bloch vector of the qubit state.

When collective measurements on two identical qubits are allowed, the precision limit is constrained by a collective bound. According to Eqs. (6.73) and (6.74) in ref. [13] with $t = 3/2$, the collective bound for the mean infidelity (mean square Bures distance) is $\frac{3}{2N}$ (cf. ref. [14]), and the bound for the MSE is

$$\begin{cases} \dfrac{(2+\sqrt{1-s^2})^2}{3N} & \text{if } 0 \le s \le \dfrac{3+4\sqrt{3}}{13}, \\[2ex] \dfrac{s(1+s)(3-s)}{(3s-1)N} & \text{if } \dfrac{3+4\sqrt{3}}{13} \le s \le 1. \end{cases} \tag{6}$$

The GM bound and collective bound for the mean infidelity may be violated when the state is nearly pure (with thresholds depending on $N$), in which case common estimators (including the maximum likelihood estimator) are biased due to the boundary of the state space. The precision limits with respect to the MSE are less sensitive to this influence.

**Data availability.** The data that support the results of this study are available from the corresponding authors upon request.

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

## Acknowledgements

The work at USTC is supported by the National Natural Science Foundation of China under Grants (Nos. 11574291, 11774334, 61327901, and 11774335), the National Key Research and Development Program of China (No.2017YFA0304100), Key Research Program of Frontier Sciences, CAS (No.QYZDY-SSW-SLH003), the Fundamental Research Funds for the Central Universities (No.WK2470000026), and China Postdoctoral Science Foundation (Grant No. 2016M602012). HZ acknowledges financial support from the Excellence Initiative of the German Federal and State Governments (ZUK 81) and the DFG. J.S. acknowledges financial support from the ERC (Consolidator Grant 683107/TempoQ), and the DFG.

## Author contributions

HZ developed the theoretical approach; GYX supervised the project; ZBH, JFT, JL, and GYX designed the experiment and the measurement apparatus for the collective measurement; ZBH built the instruments, performed the experiment and collected the data with assistance from GYX, JFT, YY, and KDW; JS developed the maximum likelihood algorithm for quantum state tomography with collective measurements. ZBH, JS, HZ, and GYX performed numerical simulations and analyzed the experimental data with assistance from CFL and GCG; HZ, ZBH, JS, and GYX prepared and wrote the manuscript.

## Additional information

**Competing interests:** The authors declare no competing interests.

