## [Peer Review File · Nature Communications]

REVIEWERS' COMMENTS:

Reviewer #1 (Remarks to the Author):

The authors have satisfactorily addressed all my previous technical comments on data statistical analysis which is now appropriate, on the scaling of the mean infidelity with mixed states, and added references to cite previous works on beam-displacing prisms quantum walk architectures. Furthermore, they have specified in which conditions collective measurements can provide an advantage for multiparameter estimation purposes.

I find that the obtained results clearly support the claims of the manuscript, and that the reported experimental data show the advantages of the implemented two-qubits SIC-POVM measurement for state tomography.

My main previous concern was related to the possibility of extending the proposed architecture to larger dimensions, namely more than two qubits or qudits with $d > 2$. I still believe that the presented platform with beam-displacing prisms interferometers cannot be easily extended to more complex scenarios, since this architecture with two qubits is limited by the polarization space dimensionality ($d=2$). Nevertheless, this experiment represents a proof-of-principle demonstration of a collective two-qubit SIC-POVM, demonstrating an advantage in employing entangled measurements in certain scenarios. Hence, I would recommend publication of this manuscript in Nature Communications.

Reviewer #2 (Remarks to the Author):

I believe the Authors have replied satisfactorily enough to my concerns and to the issues that I had raised. I also believe they replied satisfactorily to the issues raised by the other referees. I suggest the paper be published on Nature photonics.

Reviewer #3 (Remarks to the Author):

In their response, the authors have provided satisfactorily to most of the questions.

\begin{enumerate}

\item In response to Comment 2, the authors say 'the key distinction between collective measurements and separable measurements as explained in Refs. [13,14]; although multi-copy collective measurements can further improve the efficiency, the improvement is not so significant.' while in response to Comment 3, they say 'As shown in Refs. [13,14], the collective SIC-POVM is optimal among two-copy collective measurements in qubit state tomography. In other words, more entangled measurements cannot lead to better results.' Can the authors make the distinction between multi-copy measurements vis-a-vis more entangled measurements clear in their manuscript?

\item I suggest the authors add the comparison of their work to Ref.16 in their manuscript for the benefit of the readers.

\end{enumerate}

Other comments:

`\begin{enumerate}`

`\item` When I suggested that the authors avoid the use of superlative adjectives, I did not mean 'superefficient' to be replaced by 'highly efficient'. The reason is adverbs like 'highly' have no scientific meaning. The title 'Deterministic realization of collective measurements via photonic quantum walks' is a fair description of the work.

`\end{enumerate}`

In conclusion, the authors must undertake the changes mentioned above before their work can be published in Nature Communications.

0. **Comment:** The authors have satisfactorily addressed all my previous technical comments on data statistical analysis which is now appropriate, on the scaling of the mean infidelity with mixed states, and added references to cite previous works on beam-displacing prisms quantum walk architectures. Furthermore, they have specified in which conditions collective measurements can provide an advantage for multiparameter estimation purposes.

I find that the obtained results clearly support the claims of the manuscript, and that the reported experimental data show the advantages of the implemented two-qubits SIC-POVM measurement for state tomography.

My main previous concern was related to the possibility of extending the proposed architecture to larger dimensions, namely more than two qubits or qudits with $d > 2$. I still believe that the presented platform with beam-displacing prisms interferometers cannot be easily extended to more complex scenarios, since this architecture with two qubits is limited by the polarization space dimensionality ($d=2$). Nevertheless, this experiment represents a proof-of-principle demonstration of a collective two-qubit SIC-POVM, demonstrating an advantage in employing entangled measurements in certain scenarios. Hence, I would recommend publication of this manuscript in Nature Communications.

Response: Thank you very much for recommending our work for publication in Nature Communications. We agree that multi-copy collective measurements are much more difficult to implement, but we believe that our work will serve as a stepping stone and will stimulate further progresses on this topic.

0. **Comment:** I believe the Authors have replied satisfactorily enough to my concerns and to the issues that I had raised. I also believe they replied satisfactorily to the issues raised by the other referees. I suggest the paper be published on Nature photonics.

Response: Thank you very much for the appreciation of our work.

0. **Comment:** In response to Comment 2, the authors say 'the key distinction between collective measurements and separable measurements as explained in Refs. [13,14]; although multi-copy collective measurements can further improve the efficiency, the improvement is not so significant.' while in response to Comment 3, they say 'As shown in Refs. [13,14], the collective SIC-POVM is optimal among two-copy collective measurements in qubit state tomography. In other words, more entangled measurements cannot lead to better results.' Can the authors make the distinction between multi-copy measurements vis-a-vis more entangled measurements clear in their manuscript?

Response: Thank you very much for the constructive comments. To clarify this issue, we have added the following sentences at the end of the section entitled "Optimized collective measurements".

As far as two-copy collective measurements are concerned, surprisingly, more entangled measurements, such as the Bell measurements, cannot lead to higher efficiency. Although multi-copy (say three-copy) collective measurements can further improve the efficiency, the improvement is not so significant [13,14].

1. **Comment:** I suggest the authors add the comparison of their work to Ref.16 in their manuscript for the benefit of the readers.

Response: We have added the comparison of our work with Ref. [16] in the last paragraph of the main text as follows.

For example, in the joint estimation of phase and phase diffusion, it was shown in Ref. [15] that collective measurements can lead to higher precision than local measurements. Recently, this prediction was verified in a proof-of-principle experiment based on probabilistic Bell measurements [16]. Our technique for implementing deterministic collective measurements may help demonstrate the practical advantage of collective measurements in quantum metrology.

2. **Comment:** When I suggested that the authors avoid the use of superlative adjectives, I did not mean 'superefficient' to be replaced by 'highly efficient'. The reason is adverbs like 'highly' have no scientific meaning. The title 'Deterministic realization of collective measurements via photonic quantum walks' is a fair description of the work.

Response: We have deleted the phrase "highly efficient" in the title.

3. **Comment:** In conclusion, the authors must undertake the changes mentioned above before their work can be published in Nature Communications.

Response: We thank you for all the constructive suggestions and have revised the manuscript accordingly. We believe that the current manuscript is suitable for publication in *Nature Communications*.

List of changes

1. We have deleted "highly efficient" in the title according to the suggestion of Reviewer 3.
2. We have added two sentences at the end of the section entitled "Optimized collective measurements" to address the comment of Review 3 concerning more entangled measurements and multi-copy collective measurements.
3. In the last paragraph of the main text, we have added discussion on the connection between our work and Ref. [16].